# Optical Fiber Interferometric Humidity Sensor by Using Hollow Core Fiber Interacting with Gelatin Film

**DOI:** 10.3390/s22124514

**Published:** 2022-06-15

**Authors:** Yusong Zhong, Pengbai Xu, Jun Yang, Xinyong Dong

**Affiliations:** 1School of Information Engineering, Guangdong University of Technology, Guangzhou 510006, China; 2111903102@mail2.gdut.edu.cn (Y.Z.); pengbaixu@gdut.edu.cn (P.X.); yangj@gdut.edu.cn (J.Y.); 2Guangdong Provincial Key Laboratory of Information Photonics Technology, Guangzhou 510006, China

**Keywords:** optical fiber sensors, humidity measurement, gelatin, Fabry–Perot interferometer

## Abstract

An optical fiber Fabry–Perot interferometer (FPI) is constructed for relative humidity measurement by fusion splicing a short hollow core fiber (HCF) to the end of a single-mode fiber and coating the tip of the HCF with a layer of gelatin. The thickness of the gelatin film changes with ambient humidity level and modulates cavity length of the FPI. Humidity measurement is therefore realized by measuring the wavelength shift of the interreference fringe. RH sensitivity of 0.192 nm/%RH is achieved within a measurement range of 20–80%RH. Dynamic measurement shows a response and recovery time of 240 and 350 ms, respectively. Sensor performance testing shows good repeatability and stability at room temperature but also reveals slight dependence of the RH sensitivity on environmental temperature. Therefore, a fiber Bragg grating is cascaded to the FPI sensing probe to monitor temperature simultaneously with temperature sensitivity of 10 pm/°C.

## 1. Introduction

Humidity measurement is widely needed in the many fields such as food, medicines, storage and environmental monitoring. High-performance humidity sensors with wide measurement range, high accuracy and fast response speed are desired. Since conventional humidity sensors based on electronic technologies are hardly used for long-distance measurement or in strong electromagnetic-interference environments, optical fiber humidity sensors have attracted a growing number of research interests in recent years because of their long-distance signal transmission capability and immunity to electromagnetic interference.

An optical fiber humidity sensor is usually composed of an optical fiber structure interacting with a humidity-sensitive material. The optical fiber structure can be a fiber grating [1,2,3,4], fiber interferometer [5,6,7,8,9], D-shaped fiber [10,11,12], microfiber [13,14], etc. Among them, optical fiber interferometers showed relatively high sensitivity and simple structures. The reported humidity sensitive materials include polyvinyl alcohol (PVA) [15,16,17,18,19], polyimide [20,21], chitosan [22,23,24], agarose [25,26], carbon nanotubes [27], oxides [28,29,30,31], Norland optical adhesive (NOA) [32] and grapheme oxide (GO) [33,34]. Most of the above-mentioned humidity-sensitive materials were used in optical fiber interferometric humidity sensors. However, it is hard to achieve all good performances including high sensitivity, large measurement range, high response speed and good repeatability. Therefore, researchers continue to explore new sensor designs by combining new humidity-sensitive materials with various optical fiber structures for better sensing performances. It is worth noting that a gelatin-coated, four-core, fiber-based Michelson interferometric humidity sensor was recently reported and obtained a relatively high sensitivity of 0.185 nm/%RH [35]. However, the sensing structure is quite complicated and difficult to fabricate.

In this paper, an optical fiber interferometric humidity sensor by using a short hollow core fiber (HCF) interacting with a gelatin film is demonstrated. It is constructed by fusion splicing a short length of HCF to the end of a single-mode fiber (SMF) and coating the tip of the HCF with a layer of gelatin to form a Fabry–Perot interferometer (FPI). Humidity changes the thickness of the gelatin film that modulates the interference cavity length and hence the dip wavelength of the interference spectrum of the FPI in turn. A relative humidity (RH) measurement with a sensitivity of 0.192 nm/%RH is therefore achieved within a measurement range of 20–80%RH and with the response and recovery times of 240 and 350 ms, respectively. It is found that the RH sensitivity shows a slight dependence on environmental temperature. A fiber Bragg grating (FBG) is therefore cascaded in the FPI sensing probe to monitor temperature changes simultaneously.

## 2. Fabrication and Principle

A schematic diagram of the proposed optical fiber FPI humidity sensor probe is shown in Figure 1a. It was formed by the fusion splicing of an HCF with an SMF (SM-28) and coating a layer of gelatin on the tip of the HCF. The HCF has the inner and outer diameters of 75 and 125 μm, respectively. The fusion splicing was carried out by using a fusion splicer (Atomowave SFS-A60+) operated in the manual mode with the main parameter settings of overlap 15 μm, preparatory splicing fusion time 200 ms and arc discharge −25 bit. After fusion splicing, the HCF was cleaved and left only a short HCF with the wanted length. A gelatin solution of 5% concentration was then prepared by dissolving 5 g of gelatin powder into 95 mL deionized water and stirred for 30 min at temperature of 65 °C. The gelatin solution was then coated on the tip of the HCF by using the dip-coating method. It was then dried for 7 h at room temperature to form a film. The thickness of the dried gelatin film estimated from the image local difference is about 6 µm. The FBG was cascaded to the FPI sensing probe for simultaneous measurement of temperature. It was written in the SMF before the FPI fabrication by using a 248 nm excimer laser and the phase mask method. Its central wavelength, reflectivity and 3-dB bandwidth are 1562 nm, 85% and 0.2 nm, respectively. Figure 1b is an image of the proposed FPI humidity probe.

The working principle for humidity measurement is based on the humidity-induced thickness change in the gelatin film that changes the cavity length of the FPI formed by the two reflective surfaces, as shown in Figure 1a. When the light enters air hole of the HCF from the SMF, Fresnel reflection happens at the silica–air interface. When the light propagates to the gelatin film, reflection happens again at the air–gelatin interface. These two parts of the reflected light combine in the core of the SMF and lead to light interference over there. The reflected output signal contains the interference fringe of the FPI, where the dip or peak wavelengths will shift with the humidity because the latter changes the thickness of the gelatin film and hence the length of the FPI cavity.

Based on the theory of optical interference, intensity of the reflected light of the FPI probe can be derived by using the two-beam approximation and expressed as [36]:(1)$$I=I_{0}\left[R_{1}+R_{2}+2\sqrt{R_{1}R_{2}}\cos\left(\varphi \right)\right]$$
where $I_{0}$ is the intensity of the incident light, $R_{1}$ and $R_{2}$ are the reflectivity of the two interfaces, λ is the wavelength, L is the cavity length of the FPI and φ is the initial phase. The free spectral range (FSR) of the interference fringe can be expressed as:(2)$$∆\lambda =\frac{\lambda^{2}}{2nL}$$
where n is the refractive index of the air in the FPI cavity. The interference spectrum reaches its dip at the phase of resonant wavelengths satisfying φ=(2m+1)π, with m being an integer. The dip wavelengths of the interference spectrum can be expressed as:(3)$${\;\lambda }_{m}=\frac{4nL}{2m+1}\;,\left(m=1,2,3\ldots \right)$$

When the external humidity level increases, the gelatin film will absorb more water molecules from the surrounding air and expand in volume. Its thickness will be increased accordingly, which will reduce cavity length of the FPI and shift the dip wavelengths of the interference spectrum to the short wavelength direction. When the external humidity level decreases, the gelatin film will release water molecules to the surrounding air and shrink in volume. Its thickness will be decreased and the cavity length of the FPI will become larger to make the dip wavelengths of the interference spectrum move to longer wavelengths. The changes in dip wavelengths of the FPI can be expressed by:(4)$$\;{\;\Delta \lambda }_{m}=\frac{4n}{2m+1}\Delta L$$
where ΔL is the change of the FPI’s cavity length. Once we achieved the relationship of the dip wavelength shifts to RH variations, RH measurement will be obtained by measuring the interference spectrum of the optical fiber FPI probe.

## 3. Experimental Results and Discussions

The experimental setup of the humidity measurement is shown in Figure 2. It consists of a broadband light source (BBS), an optical spectrum analyzer (OSA, MS9740A), an optical fiber circulator (OC), the FPI probe and a temperature- and humidity-controlled chamber (THCC). The FPI probe was mounted to a fixture to avoid fiber bending and the effect of vibrations and then placed into the THCC for testing. The temperature was maintained firstly at 20 °C, and the RH was changed at a step of 5%RH within the range of 20 to 80%RH. After each change, we waited for over 10 min for the purpose of the film to reach its equilibrium at the new RH level and then recorded the spectrum and read the dip wavelengths.

Three FPI probes with different HCF lengths of 73, 110 and 220 μm were fabricated, and their interference spectra are shown in Figure 3. Their FSRs are 17.10, 10.19 and 5.25 nm, respectively. It decreases with the HCF length or cavity length, as predicted by Equation (2). However, the contrast of the interference fringe decreases with the HCF length, too. It is 28.96, 14.89 and 9.28 dB for the HCF lengths of 73, 110 and 220 μm, respectively. The increasing transmission loss of the light signal with cavity length should be the main reason for the contrast reduction. In the following experiments of humidity measurement, we used the FPI probe with the smallest HCF length of 73 μm.

The humidity level was increased at a step of 5%RH within the range of 20 to 80 %RH. Figure 4 shows the measured reflection spectra of the FPI probe under different RH levels of 20, 40, 60 and 80%RH. A significant wavelength shift of 11.53 nm towards the short wavelength direction was observed for the interference dip when RH was increased from 20 to 80%RH. The reflection peak wavelength of the FBG had nearly no changes, indicating that the FBG is not sensitive to RH at all.

We tested the responses of the FPI probe in both ascending and descending orders of RH level. The measured wavelength shifts in the FPI probe against RH in both orders as shown in Figure 5 are found to be very close to each other, indicating that the proposed FPI probe has good reversibility. The calculated RH sensitivity of the FPI probe by linear fitting is 192 pm/%RH. The measured wavelength shifts in the FBG’s reflection peak are also shown in Figure 5. The maximum variation is only 10 pm, less than the wavelength resolution of the OSA.

To further study the repetitive characteristics of the FPI humidity probe, two more iterations of repetitive testing were carried out. The measured results of dip wavelength shift against RH level in both ascending and descending orders are shown in Figure 6. The mean square deviation of the six sets of data is 3.96 × 10^−2^, indicating quite good repeatability for the proposed FPI humidity probe. The minor errors among different measurements are most possibly caused by the nonuniform RH distribution and unstable airflows in the THCC.

To study the effect of environmental temperature on the response of the FPI humidity probe, the testing was carried out under two more different temperatures of 15 and 25 °C. The measured results, together with that measured at 20 °C, were shown in Figure 7. It can be seen that all the three serials of the wavelength shift of the FPI probe have good linear relationships with the RH level, and the calculated humidity sensitivities by linear fitting are 173, 192 and 194 pm/%RH for temperatures of 15, 20 and 25 °C, respectively. The sensitivity increases slightly with temperature, and the change is relatively larger when the temperature was changed from 15 to 20 °C than that from 20 to 25 °C. It is known that more water molecules can be contained in the air if the temperature increases, so the gelatin film of the FPI probe may absorb more water molecules under a higher temperature, especially when the relative humidity is high. That may be the reason why the humidity sensitivity increases with temperature.

Then, the stability test of the FPI probe against time was carried out at three different RH levels of 20, 40 and 80%RH under a fixed temperature of 20 °C. The dip wavelength was measured every 5 min within a period of 30 min, and the results are shown in Figure 8. It can be seen that the measured wavelength at each RH value is quite stable within the checking period. The recorded fluctuations are no more than ±0.28 nm in the three RH cases. Taking the 194 pm/%RH sensitivity into account, the maximum error corresponds to 1.44%RH only, which can be neglected in most applications. The small fluctuation in the measurement results may also be related to the nonuniform RH distribution and unstable airflows in the THCC.

The dynamic response of the FPI probe to humidity change was then carried out by changing the humidity level from 50 to 80%RH quickly and then changing it back after 10 s. We set the OSA to automatic acquisition mode to record the wavelength shift of interference spectrum of the FPI probe at a high speed. The measured results, as shown in Figure 9, indicating that the response and recovery time, which represent the time taken for the sensor to reach 90% from 10% of its final value and for the reverse process, are about 240 and 350 ms, respectively. It should be mentioned that the humidity switching time was also counted in, so the actual dynamic response and recovery time should be much smaller. However, they are much faster than those reported in previous references, as shown in Table 1.

The fast response and recovery time may be attributed to the compact structure of the FPI probe and the small thickness of the gelatin film, which allow the fast diffusion of water vapor molecules. Based on the fast response time, the FPI probe provides a near-real-time humidity measurement technique which can be used in areas such as human breathing monitoring for the improvement of healthcare, as the response time of current commercial humidity sensors is in the order of several seconds.

Finally, we tested the sensitivity of the FBG to temperature within a temperature range of 10 to 50 °C. The measured wavelength of the FBG’s reflection peak against temperature are shown in Figure 10. The calculated temperature sensitivity of the FBG sensor is 10 pm/°C with high linearity R^2^ = 0.99723. Since the FBG is not sensitive to RH variations, it can provide the accurate information of environmental temperature for reference when the FPI humidity probe works.

To better evaluate the proposed FPI humidity sensor, we compare the key performances including sensitivity, measurement range and dynamic response time of our sensor, with other reported ones also based on optical fiber interferometers. The results, as shown in Table 1, indicate that our humidity sensor is among the best in all the three listed parameters of humidity-sensing performance. Combined with other merits of low cost, linear response, high stability and the capability of simultaneous measurement of temperature, the FPI humidity sensor may have broad application prospects in related industry areas.

## 4. Conclusions

An optical fiber FPI-based humidity sensor has been demonstrated by fusion splicing a short HCF with a single-mode fiber and coating the tip of the HCF with a layer of gelatin. The working principle is based on the humidity-induced thickness change in the gelatin film that changes the cavity length of the FPI. A humidity measurement with a high sensitivity of 0.192 nm/%RH and a wide measurement range of 20–80%RH has been realized by measuring the wavelength shift of the interreference fringe. Good repeatability and stability have been obtained, and the response and recovery times are only 240 and 350 ms, respectively. In addition, simultaneous measurement of temperature with a sensitivity of 10 pm/°C has been achieved by cascading an FBG sensor in the FPI probe.

## Figures and Tables

**Figure 1 sensors-22-04514-f001:**
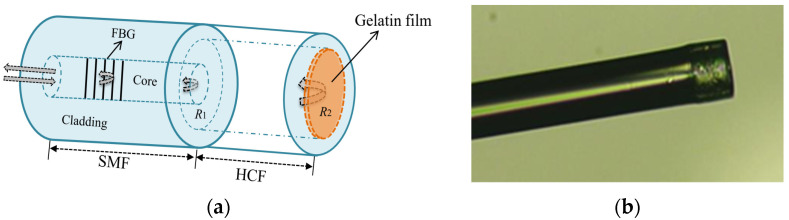
(**a**) Schematic diagram and (**b**) image of the proposed FPI humidity probe.

**Figure 2 sensors-22-04514-f002:**
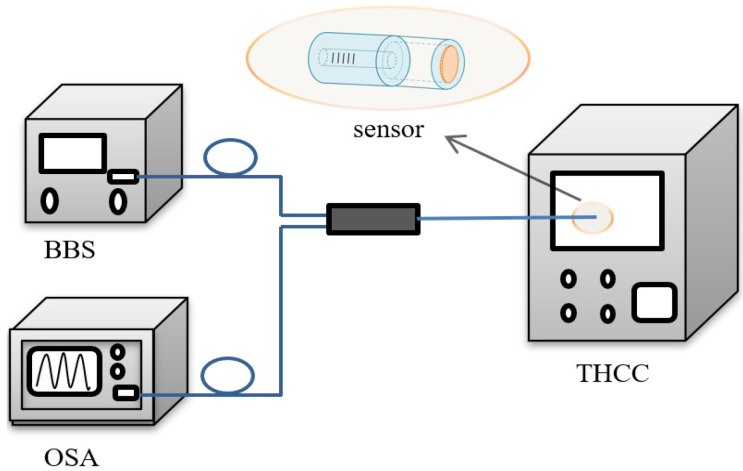
Experimental setup of the measuring system.

**Figure 3 sensors-22-04514-f003:**
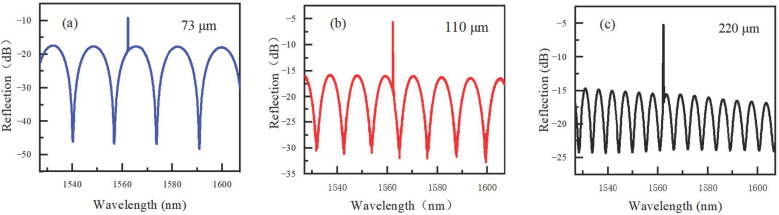
Spectra of the FPI probe with different hollow core fiber lengths of (**a**) 73, (**b**) 110 and (**c**) 220 μm.

**Figure 4 sensors-22-04514-f004:**
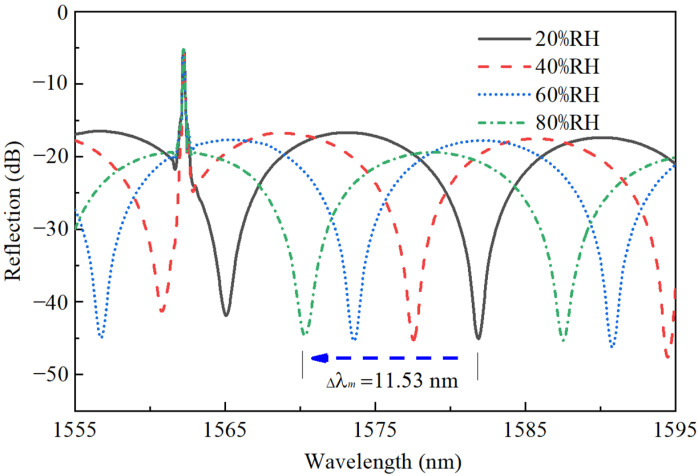
Spectra of the FPI probe at different relative humidity.

**Figure 5 sensors-22-04514-f005:**
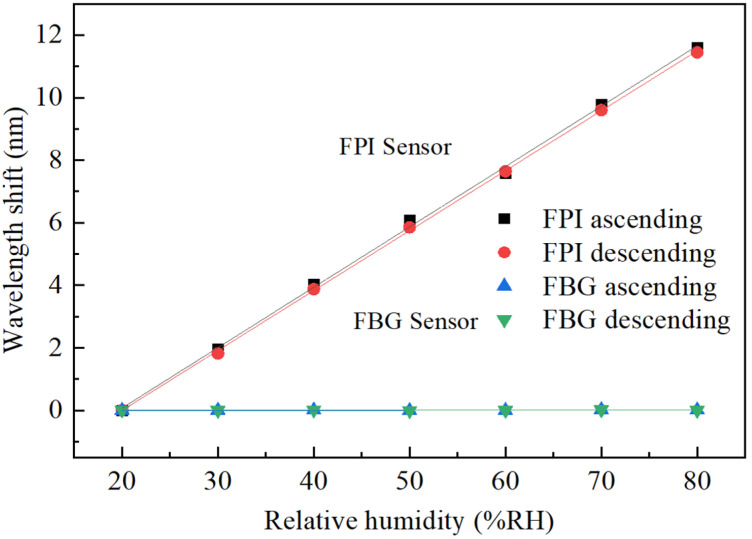
Wavelength shift against relative humidity.

**Figure 6 sensors-22-04514-f006:**
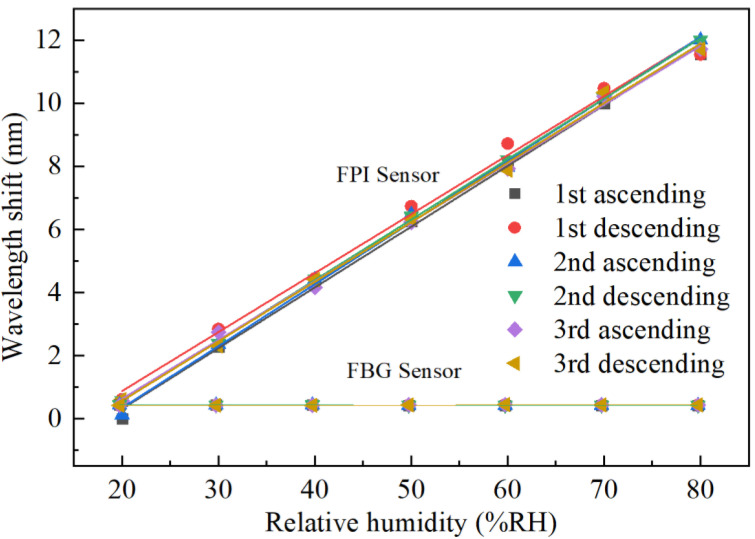
Wavelength shift against RH for three times repetitive measurements of the FPI probe.

**Figure 7 sensors-22-04514-f007:**
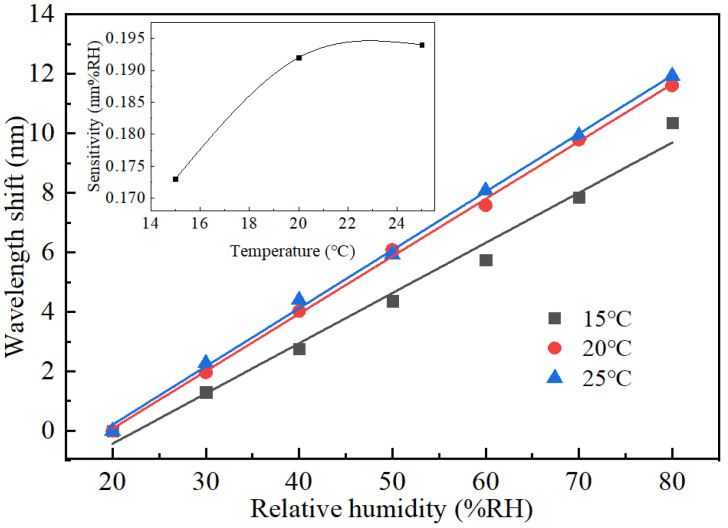
Wavelength shift against RH at different temperatures. Insert is humidity sensitivity of the FPI probe against temperature.

**Figure 8 sensors-22-04514-f008:**
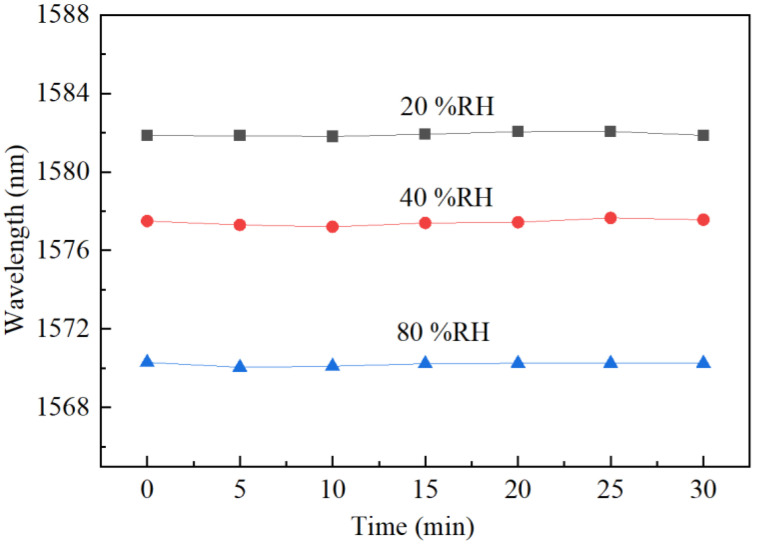
Stability test results of the FPI probe at different RH levels over 30 min.

**Figure 9 sensors-22-04514-f009:**
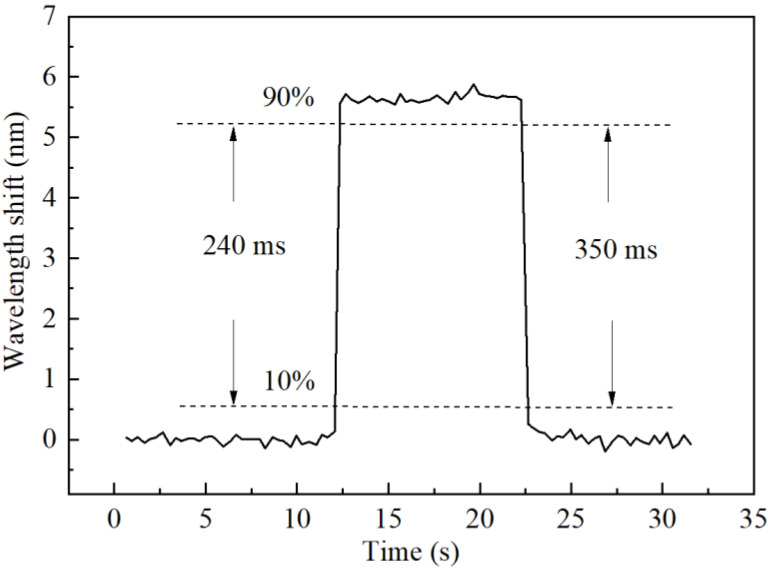
Dynamic performances of the FPI probe.

**Figure 10 sensors-22-04514-f010:**
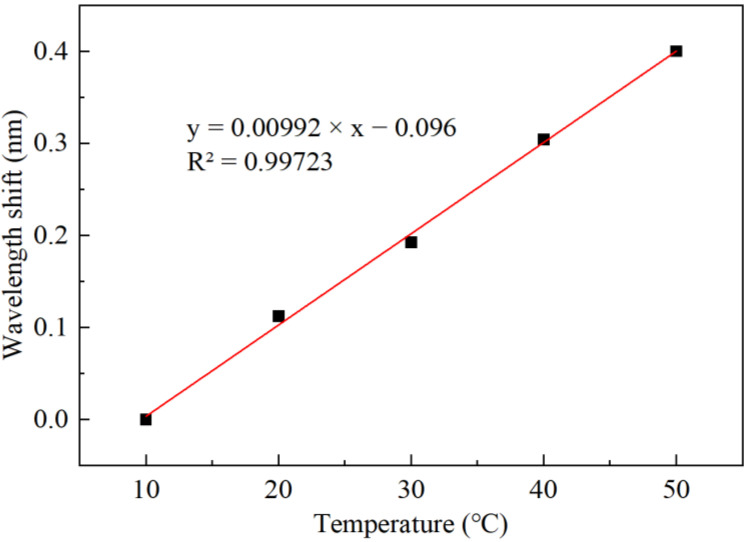
Variation in the FBG sensor’s wavelength with temperature.

**Table 1 sensors-22-04514-t001:** Performance comparison of some reported optical fiber humidity sensors.

Type	Measurement Range (%RH)	Sensitivity	Response/Recovery	Reference
MZI + GO	80–90	−0.885 dB/%RH	0.42/6.54 s	[9]
MI + GO	60–90	0.22 dB/%RH	5.2/8.1 s	[7]
MZI + GQDs-PVA	13.77–77.87	−0.0901 nm/%RH	NA	[5]
FPI + GQDs-PVA	13.47–81.34	117.25 pm/%RH	NA	[6]
FPI + NOA	20–90	0.0545 nm/%RH	5 s	[30]
FPI +PVA	7–91.2	0.07 nm/%RH	NA	[18]
FPI + Ti_3_O_5_/SiO_2_	1.8–74.7	0.43 nm/% RH	5/5 s	[31]
FPI + Chitosan	20–95	0.13 nm/%RH	380 ms	[21]
FPI +PDMS/PVA	20–45	0.128 nm/%RH	NA	[19]
FPI + Polyimide	20–90	0.022 nm/%RH	NA	[20]
MI + Chitosan	57.3–83.5	135 pm/%RH	5/3 s	[22]
MI + Gelatin	45.0–81.7	−0.185 nm/%RH	5.24/7.12 s	[33]
FPI + Gelatin	20–80	192 pm/%RH	240/350 ms	Our work

## Data Availability

Not applicable.

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
