# Peer review of "Optical Fiber Interferometric Humidity Sensor by Using Hollow Core Fiber Interacting with Gelatin Film"

_sensors, 2022, doi:10.3390/s22124514_

Round 1
Reviewer 1 Report
This manuscript presents an optical fiber Fabry-Perot interferometer for relative humidity measurement. I can recommend this manuscript for publication only in the revised form.
Comments: 1. The principle of operation of the device is not explained.2. Nothing is clear from Figure 1 Figure needs to be improved
3. Eq. (1) needs to be explained. It is desirable to present its derivation.
Reviewer 2 Report
In this work, Zhong et al., fabricated an optical fiber Fabry-Perot interferometer (FPI) by fusion splicing a short hollow-core fiber (HCF) to the end of a single-mode fiber and coating the tip of the HCF with a layer of gelatin. They have used it for relative humidity measurements. The obtained results are reasonable and the manuscript is well organized. Hence, I recommend this work for publication in Sensors but the manuscript should be checked properly for grammatical errors. The authors are advised to polish the English, especially Abstract, Introduction and Conclusion.
Reviewer 3 Report
The authors present an optical fiber interferometric humidity sensor by using hollow-core fiber interacting with gelatin film. The excellent performance of the sensor in terms of measurement range, sensitivity, and speed could be of interest to the community. However, the manuscript does not have discussions about their results even in the Discussion section. I would suggest the authors address this before publication.
Reviewer 4 Report
The manuscript entitled, "Optical fiber interferometric humidity sensor by using hollow 2 core fiber interacting with gelatin film" submitted by Zhong et al represents a nice study using gelatin as humidity sensitive film deposited at the end of hollow core fiber. The results are fairly good and presented nicely. However, it lacks many points:
1. A broad range of work has been done using either chitosan or gelatin over FP interferometry based fiber optic probe so novelty is missing hugely.
2. The details of HCF are missing and look slightly misleading from the schematic shown in the figure which represents the fiber with a large air core and thin cladding. It is better to present the optical microscopic image.
3. What is gelatin film thickness and how does it affect the sensitivity?
4. Figure 7: Authors should also provide a relation between humidity sensitivity with respect to ambient temperature for better calibration.
5. Line 176: "changed it back after 10 minutes" should be "changed it back after 10 seconds" as seemed from the figure.
6. Figure 9: What is the response time of the humidity chamber, which should also be taken into consideration during measurements.
7. Table: A number of papers are still missing for comparison, such as :
Optical Fiber Technology, 20(4), 314-319.
Scientific reports, 10(1), 1-10.
even more.......
Based on these comments, I recommend a major revision of the manuscript at the current stage.
Round 2
Reviewer 4 Report
Authors have successfully addressed all the comments. MS can be published in the current form